## Replications

psychology

terror management theory, mortality salience, meta-analysis, replication, publication bias

**Author for correspondence:**
Miguel A. Vadillo
e-mail: miguel.vadillo@uam.es

# Are we truly special and unique? A replication of Goldenberg *et al.* (2001)

Javier Rodríguez-Ferreiro[1,2], Itxaso Barberia[1], Jordi González-Guerra[1] and Miguel A. Vadillo[3]

[1]Departament de Cognició, Desenvolupament i Psicologia de l'Educació, and [2]Institut de Neurociències, Universitat de Barcelona, Barcelona, Spain
[3]Departamento de Psicología Básica, Universidad Autónoma de Madrid, Madrid, Spain

(iD) MAV, 0000-0001-8421-816X

According to the mortality salience hypothesis of terror management theory, reminders of our future death increase the necessity to validate our cultural worldview and to enhance our self-esteem. In Experiment 2 of the study 'I am not an animal: Mortality salience, disgust, and the denial of human creatureliness', Goldenberg *et al.* (Goldenberg *et al.* 2001 *J. Exp. Psychol. Gen.* **130**, 427–435. (doi:10.1037/0096-3445.130.3.427)) observed that participants primed with questions about their death provided more positive evaluations to an essay describing humans as distinct from animals than control participants presented with questions regarding another aversive situation. In a replication of this experiment conducted with 128 volunteers, we did not observe evidence for a mortality salience effect.

## 1. Introduction

Terror management theory (TMT; [1,2]) posits that humans need to reconcile their animal self-preservation instinct with the awareness that they will sooner or later die. To cope with the 'terror' produced by this paradox, people activate a dual-component *cultural anxiety-buffer* based on (i) the faith in one's cultural worldview, which '… imbues life with meaning, order, and permanence, and the promise of safety and death transcendence to those who meet the prescribed standards of value' ([2], p. 71) and (ii) self-esteem, that is, the conviction that one is living according to the standard of value of that cultural worldview.

One of the most important corollaries of TMT is the mortality salience (MS) hypothesis [3]. According to this hypothesis, if the contemplation of our own death produces terror, reminders of future death should have the potential to increase the necessity of protecting our cultural worldview and validating our self-esteem [2]. In a standard experiment, MS is manipulated, for instance, by

**Figure 1.** Reanalysis of Burke *et al.* [3]. (*a*) Funnel plot of effect sizes against their standard errors. Data points inside the grey triangle are non-significant in a two-tailed test with $\alpha = 0.05$. The red line represents Egger's regression test for funnel plot asymmetry. (*b*) Distribution of the ratio between each effect size and its standard error. Values larger than 1.96 (denoted by the red vertical line) are statistically significant in a two-tailed test with $\alpha = 0.05$.

asking participants in the experimental group to answer two open questions that force them to think about their own death, whereas participants in the control group are usually asked analogous questions about a non-death-related topic (e.g. [4,5]). After the MS manipulation, and usually following a delay (e.g. [6]), the activation of the cultural anxiety-buffer is assessed.

This activation has been measured in many different ways. Just to cite a few examples, research has shown that participants exposed to a MS manipulation increase their relative preference for positive opinions over negative opinions about their country (e.g. [4,7]), tend to give a larger amount of hot sauce to a person who disagrees with their worldview [8], evaluate more positively an essay that presents humans as unique and different from other animals [9], and report a stronger desire for offspring [10].

A systematic review and meta-analysis published almost a decade ago by Burke *et al.* [3], identified 277 MS studies and estimated that their average effect size was a medium-to-large effect of $r = 0.35$, 95% CI [0.32, 0.37]. According to Burke *et al.* [3], the distribution of effect sizes in this literature did not reveal any evidence of publication or reporting biases, suggesting that the effect was robust and reliable. However, contrary to this conclusion, a reanalysis of the same data conducted just a few years later by Yen & Cheng [11] did find signs of bias, at least within a set of studies conducted by a specific team of American researchers, who systematically reported above-average effect sizes.

Our own inspection of the data reported by Burke *et al.* [3] converges to the conclusion that the robustness of these effect sizes may have been overestimated. In figure 1*a* we replot the distribution of effect sizes reported by Burke *et al.* against their standard errors. Instead of analysing raw correlation coefficients, which are constrained to adopt values from −1 to +1 and do not follow a normal distribution, we converted all effect sizes to Fisher's *z* scores. In the absence of publication or reporting biases, the distribution of effect sizes should be symmetric. However, the selective publication of studies (or analyses) with significant results can induce a positive relation between effect sizes and standard errors, because studies with larger standard errors (i.e. smaller sample sizes) require larger effects to reach statistical significance.

Egger's regression test, represented as a red line in figure 1*a*, confirms that the distribution of effect sizes is significantly asymmetric, $z = 2.06$, $p = 0.039$, suggesting that these effect sizes may be inflated by the selective publication of significant results. This asymmetry seems to be due to the fact that a large proportion of effect sizes tend to be just slightly larger than the border of statistical non-significance, represented by the grey triangle in figure 1*a*. This is perhaps easier to appreciate in figure 1*b*, showing that the modal value of the ratios between each effect size and its corresponding standard error is just slightly above 1.96, i.e. the threshold for statistical significance in a two-tailed test. Additionally, a

selection model that assumes publication bias [12] fits the data significantly better than a traditional random-effects meta-analysis, $\chi_1^2 = 47.87$, $p < 0.001$, and returns a bias-corrected average effect size of $r = 0.22$, 95% CI [0.14, 0.30]. Although significant, this effect is 37% smaller than the meta-analytic effect reported by Burke *et al.* [3].

Of course, the previous analyses do not prove conclusively that these studies are influenced by publication or reporting biases, as funnel-plot asymmetry can arise for reasons that have nothing to do with bias of any kind [13]. And even if we could be completely sure that this set of studies is biased, that would not imply by any means that the effects explored in this literature are trivial or nonexistent, although it would suggest that the effect sizes of the published record possibly overestimate the true effects of MS manipulations. In this context, replication of key experimental findings from this literature can be a valuable means to assess the reliability of the main results and obtain a realistic estimate of their effect sizes.

With this in mind we decided to replicate one of the studies from the main authors of this theory ([9], Study 2). In this study, participants primed with questions about their death provided more positive evaluations to an essay describing humans as distinct from animals than control participants presented with questions regarding another aversive situation. There are several reasons why we considered this study ideally suited for a replication attempt. Many MS studies rely on culturally laden materials that cannot be easily adapted to other populations, such as evaluating anti-American essays or measuring attitudes towards specific minorities or foreigners (for a review, see [14]). In contrast, the materials of the study conducted by Goldenberg *et al.* [9], described below, can be straightforwardly applied to any Western European population. In addition, the effect size for this study was considerably larger than average in the meta-analysis conducted by Burke *et al.* [3], which, in principle, should improve the reproducibility of the effect. Furthermore, with more than 360 citations in Google Scholar at the time of writing the present article, Goldenberg *et al.* [9] is one of the most influential studies reviewed by Burke *et al.* [3]. In the original study, an essay describing humans as similar to other animals ('humans are animals' essay) was also used. Nevertheless, differences between the experimental and control group were observed only in relation to the 'humans are unique' essay. Therefore, only this essay is studied in our replication.

## 2. Methods

### 2.1. Participants

According to the information provided in table 2 of Goldenberg *et al.* [9] the effect size of the key statistical contrast we intended to replicate is $d = 1.13$. A power analysis with G*Power [15] showed that only 44 participants (22 per group) were needed to detect an effect of this magnitude with 0.95 power in a two-tailed *t*-test for independent samples. Therefore, we planned to test a group of at least 44 participants, but we left open the possibility of testing more participants if they were available in our campus. In practice, we expected that the participant recruiting system would allow us to test at least 100 participants.

The final sample consisted of 128 psychology undergraduate students from the University of Barcelona (110 female), who took part in the experiment in exchange for course credit (mean age = 19.9, s.d. = 2.17). The participants were randomly assigned to the experimental or control conditions. The mean ages of both groups were not significantly different, $t_{126} = -1.062$, $p = 0.209$. All the participants were native Spanish-speakers.

### 2.2. Procedure

The present study replicated the procedure followed in Study 2 by Goldenberg *et al.* [9]. Participants were tested in groups of up to 10 volunteers. Upon arrival, they were informed that they were going to take part in two short studies presented as Study 1 and Study 2. They were handed the materials for the two studies stapled together and were instructed to wait for further instructions when they finished the first study.

The first part of the experiment (Study 1), presented as a personality and attitudes assessment, started with all the filler questionnaires presented in order: Rosenberg Self-Esteem Scale ([16], Spanish version by [17]), the neuroticism scale of Eysenck's Personality Questionnaire [18], Self-Objectification Questionnaire [19], Body Self-Esteem Scale [20], followed by the MS manipulation. As in the original study, MS was manipulated with two open questions about one's own death for the experimental condition (i.e. 'Please briefly describe the emotions that the thought of your own death arouses in you' and 'Jot down, as

specifically as you can, what you think will happen to you as you physically die and once you are physically dead') and two similar questions about dental pain for the control condition ('Please briefly describe the emotions that the thought of dental pain arouses in you' and 'Jot down, as specifically as you can, what you think will happen to you next time you suffer dental pain and once the pain is over'). The Spanish translation of these questions and accompanying text used in the experiment, together with the original materials in English, are available at https://osf.io/f9ya7/. The Spanish version of the questions used in the experimental condition was taken verbatim from Barberia et al. [21].

Study 1 finished with participants completing the Positive and Negative Affect Scale (PANAS; [22]; Spanish translation of items by [23,24]). We used the version of the PANAS which instructs participants to refer to their current affective state. This scale was included in the original study in order to assess possible effects of MS on affect and to introduce a delay between MS manipulation and worldview assessment.

As in Goldenberg et al. [9], the experimenter introduced Study 2 after all the participants in each group had finished Study 1. In this second part of the experiment, the participants were asked to read and provide their opinion regarding an essay allegedly written by an honours student from a Catalan university earlier in the semester. We translated the original Goldenberg et al.'s [9] essay describing humans as different to other animals ('humans are unique' essay). The essay was followed by six evaluation questions about the author (see [9], p. 432). All questions were answered on 9-point scales where 1 corresponded to the most negative evaluations and 9 to the most positive evaluation. The Spanish translation of the essay and the six questions, together with the original materials in English, are available at https://osf.io/f9ya7/. Finally, participants were asked to describe what they believed the purpose of the study was.

For exploratory purposes we also measured the waiting time of each participant (from the completion of Study 1 to the beginning of Study 2). We report all measures, manipulations and exclusions applied in the study.

# 3. Results

The full dataset of the present study is publicly available at https://osf.io/f9ya7/. None of the volunteers discovered the true purpose of the experiment or thought that the two studies were connected to each other. Participants' responses to each of the six questions in both groups are presented in table 1. As in the original study, the overall reaction to the essay was computed averaging the responses to the six questions. Cronbach's $\alpha$ for the questionnaire was 0.94. As can be seen in figure 2, reactions to the essay were similar in the two conditions (mortality salience: $M = 5.84$, s.d. $= 1.78$; dental pain: $M = 5.67$, s.d. $= 1.98$; $t_{126} = 0.512$, $p = 0.610$). The observed effect size for this contrast, $d = 0.09$, 95% CI [$-0.26$, 0.44] was significantly different from the effect size of the original study, $d = 1.13$, 95% CI [0.17, 2.07], $z = 2.03$, $p = 0.043$. There were no significant differences in relation to their positive (mortality salience: $M = 3.02$, s.d. $= 0.79$; dental pain: $M = 2.77$, s.d. $= 0.74$; $t_{126} = 1.834$, $p = 0.069$, $d = 0.32$) or negative affect (mortality salience: $M = 1.56$, s.d. $= 0.62$; dental pain: $M = 1.46$, s.d. $= 0.50$; $t_{126} = -1.040$, $p = 0.301$, $d = 0.18$).

To quantify the extent to which the present results disagree or converge with those of Goldenberg et al. [9], we computed the 'replication Bayes factor' [25] using the statistical package JASP. For this analysis, we used a prior distribution informed by Goldenberg et al.'s original result. Specifically, we used a normal distribution with mean $d = 1.13$ and standard error 0.482. The analysis yielded a Bayes factor of 19.79 in favour of the null hypothesis. Following the suggestion of Dienes [26], we conducted a second Bayes factor analysis modelling the alternative hypothesis as a half-normal distribution with the standard deviation set to the raw effect size of the original study (i.e. 1.3 Likert units). This approach provides a simple means to implement the intuition that smaller effects are more likely than larger effects. This alternative analysis returned a Bayes Factor of 2.63 in favour of the null hypothesis.

# 4. Discussion

Overall, the pattern of results reported in the previous section are inconsistent with the original results of Goldenberg et al. [9]. This does not necessarily mean that the original effect was a false positive, but it does suggest that (i) the effect may be substantially smaller than originally reported or (ii) that the effect is extremely sensitive to contextual factors and perhaps absent in some populations. Consistent with the former interpretation, our own reanalysis of Burke et al. ([3]; see also [11]) provides compelling evidence that the effect sizes of previous research on mortality salience may have been overestimated.

But, of course, it is possible that our experiment simply failed to recreate the ideal conditions for the emergence of the mortality salience effect reported by Goldenberg et al. [9]. Failed replications of

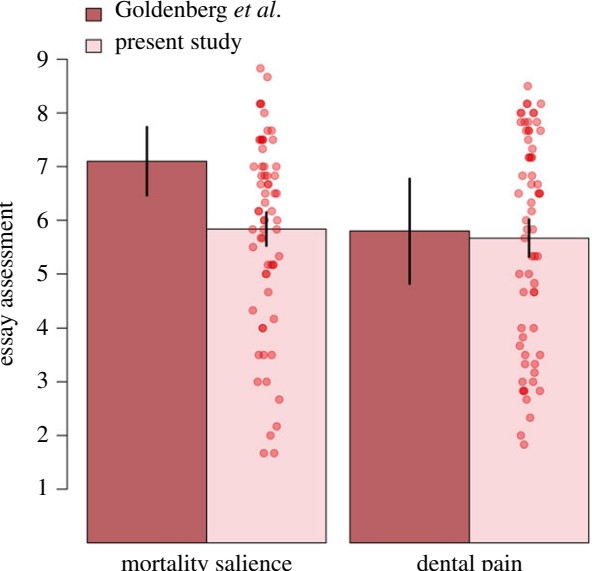

**Figure 2.** Results of the present study. Each red point represents the average essay evaluation from an individual participant. The darker bars summarize the results of the original study by Goldenberg *et al.* ([9], Study 2), whereas the lighter bars summarize the results of the present study. Error bars denote 95% confidence intervals.

**Table 1.** Item-level comparison of both groups.

| item | mortality salience | | dental pain | | *t* | *p* | *d* |
|---|---|---|---|---|---|---|---|
| | mean | s.d. | mean | s.d. | | | |
| How much do you think you would like this person? | 6.08 | 1.78 | 5.78 | 2.00 | 0.89 | 0.377 | 0.16 |
| How intelligent do you believe this person to be? | 6.39 | 1.42 | 6.27 | 1.62 | 0.46 | 0.643 | 0.08 |
| How knowledgeable do you believe this person to be? | 5.53 | 2.11 | 5.55 | 2.27 | −0.04 | 0.968 | −0.01 |
| Is this person's opinion well-informed? | 5.47 | 2.26 | 5.34 | 2.36 | 0.31 | 0.760 | 0.05 |
| How much do you agree with this person's opinion? | 5.80 | 2.36 | 5.44 | 2.65 | 0.81 | 0.420 | 0.14 |
| From your perspective, how true do you think this person's opinion is of the topic they discussed? | 5.75 | 2.28 | 5.62 | 2.37 | 0.30 | 0.762 | 0.05 |

prominent social psychology studies have often been attributed to the contextual sensitivity of the processes involved in these effects (e.g. [27–29]). For instance, reviewers of an earlier version of this article suggested that perhaps Spanish participants (i) do not care so much about uniqueness and free will, (ii) do not care about distancing themselves as much from animals as Americans, (iii) do not fear death as much as Americans. Although none of these possibilities can be discarded conclusively on the basis of the present data, we deem them unlikely. Visual inspection of figure 2 shows that the distribution of the ratings provided by our participants are in almost perfect agreement with the ratings provided by the control group in the original study by Goldenberg *et al.* ($M = 5.80$, s.d. = 1.36). This provides little support to the idea that our essays did not resonate with these participants in the same manner as they did with participants in the original study.

Of course, our study is not without limitations. Neither our empirical study nor our reanalysis of the data reported by Burke et al. [3] were preregistered, failing to meet the highest standards of confirmatory research. In contrast, we do offer public access to our complete dataset and invite sceptical readers to test alternative hypotheses that we may have overlooked and that perhaps may provide stronger support for the mortality salience hypothesis. Similarly, although the sample size recruited for the present study ($N = 128$) was substantially larger than the sample size of the original study ($N = 20$, in the same two conditions), given the evidence of bias that we detected in the literature, it might have been wise to power our study for a substantially smaller effect size, perhaps around $r = 0.22$, for consistency with the bias-corrected average effect returned by the selection model.

In any case, we think that given the theoretical relevance of this effect, it is worth investing more time and resources in establishing its reliability and boundary conditions. We hope that the present work will provide some initial momentum for further replication studies on terror management theory and the mortality salience hypothesis.

## Open practices statement

This article received results-blind in-principle acceptance (IPA) at Royal Society Open Science. Following IPA, the accepted Stage 1 version of the manuscript, not including results and discussion, was preregistered on the OSF (https://osf.io/jb7ae). This preregistration was performed after data analysis. The materials and data are publicly available at https://osf.io/f9ya7/.

Ethics. We obtained written informed consent from all the volunteers prior to their participation. Data were gathered and analysed anonymously. The study protocols were approved by the university's ethics committee (IR800003099).
Data accessibility. The full dataset of the present study is publicly available at https://osf.io/f9ya7/.
Authors' contributions. All authors made substantial contributions to this paper. All authors contributed to the design of the study. M.A.V. reanalysed the data from previous meta-analyses. All authors contributed to the writing of the paper and its revision, and approved the final version for publication.
Competing interests. We declare we have no competing interest.
Funding. J.R.-F., I.B. and M.A.V. were supported by grant nos. PSI2016-80061-R (AEI/FEDER UE), PSI2016-75776-R (AEI/FEDER UE) and PSI2017-85159-P (AEI/FEDER UE), respectively. M.A.V. was also supported by grant no. 2016-T1/SOC-1395 (Comunidad de Madrid; Programa de Atracción de Talento Investigador).

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
