## [Reviewer comments · Royal Society Open Science]

Review History

RSOS-191114.R0 (Original submission)

Review form: Reviewer 1

Do you have any ethical concerns with this paper?

No

Have you any concerns about statistical analyses in this paper?

No

Recommendation?

Accept in principle

Comments to the Author(s)

The authors propose to conduct a replication of a 2001 study about the mortality salience hypothesis. The original study found that a reminder of one's mortality produced a huge effect on subjects' evaluations of an essay emphasizing the distinction between humans and other animals, in keeping with the mortality-salience hypothesis.

My invitation to review this manuscript noted that the review process includes two stages. In stage 1, reviewers evaluate a manuscript that excludes the results, and, in stage 2, reviewers evaluate the completed manuscript. Notably, the journal explicitly allows for the submission of replication studies that have already been completed as long as the results are omitted from the Stage 1 manuscript.

I recommend that this replication proposal be accepted. The manuscript is well-written and clear, and the original study is sufficiently important (e.g., it appeared in top-tier journal and has been cited several hundred times). I have some minor comments and recommendations, which I'll include in my responses to the five review criteria listed in instructions for reviewers.

Primary Criterion #1: Whether the authors provide a sufficiently clear and detailed description of the methods ...

No concerns.

Primary Criterion #2: Whether the manuscript describes a sufficiently valid (i.e. close) and robust (e.g. statistically powerful) replication...

As for closeness of replication, the principal difference between the original study and replication study, as the authors note, is native language (original study in Colorado vs. replication study in Barcelona). Despite the salience of this difference, I don't think it's an issue because I cannot see why native language would moderate an effect related to one's sense of mortality. In fact, the two samples are not that different, as both samples consist of college students from a Western democracy consisting primarily of people who report beliefs aligning with Judeo-Christian principles. So, although I would ask that the authors list the English vs. Spanish difference in the final paper, I cannot see who anyone can argue that this difference explains why such a large effect would disappear (assuming that the effect disappears in the replication study).

As for statistical power, the intended sample size is only 44 subjects (22 per group), though the authors add that they will test more subjects if possible. The authors' power analysis indicates that 44 subjects provides statistical power of .95, but this analysis assumed an effect size of $d = 1.13$, which was observed in the original study. (The original study included 41 subjects.) Although the power calculation is appropriate, I think 44 is too few IF the replication study obtains an effect size that is greater than, say, $d = 0.20$. But if the effect size is trivial OR if the sample size is much larger than 44, I see no problems.

C. Secondary Criterion #1: The logic, rationale, and plausibility of the proposed hypotheses.

No concerns.

D. Secondary Criterion #2: The soundness of the methodology and analysis pipeline.

I would ask that the authors compare their obtained effect size to the original effect size (e.g., 95% confidence intervals).

E. Secondary Criterion #3: Whether the authors have considered sufficient outcome-neutral conditions (e.g. absence of floor or ceiling effects; positive controls; other quality checks)...

The possibility of floor or ceiling effects seems worthy of consideration. For that reason, perhaps the authors should report the M and SD for each of the six 9-point Likert questions that comprise the dependent measure (even though this was not done in the original study).

Review form: Reviewer 2

Do you have any ethical concerns with this paper?

No

Have you any concerns about statistical analyses in this paper?

I do not feel qualified to assess the statistics

Recommendation?

Reject

Comments to the Author(s)

Review of "We are all Animals: A Failure to Replicate Goldenberg et al. (2001)"

To be frank, I am not able to adequately critique the first part of the analysis and comment on whether a publication bias may have influenced the effect size of the reviewed studies. I do not have the appropriate expertise to critique the analysis. I hope you will consult additional reviewers with respect to that issue.

But it does not seem clear to me how the efforts to replicate one study address the question about publication bias. Yet, it seems that is what the authors are intending to imply.

I also question the authors' decision with regard to the study they selected to test the mortality salience effect, and the lack of consideration as to how the different cultural context and translation of content could affect the likelihood of replication. If the question is whether the tendency of mortality salience to promote efforts to protect one's worldview or self-esteem is in question, it seems important to determine that the worldview or self-esteem constituent is applicable to the population from which the sample is drawn.

Goldenberg et al. explicitly discuss the question of universality. They state that "even granting the proposition that all worldviews help people manage their terror of death, there may be cultural differences in the need to distance from animals." So really, the conclusion that can be drawn here is not that the results cast doubt on the reliability of mortality salience effects, but rather that the need to distance from animals as a means to defend against mortality salience may or may not be universal. But, it also may be that the essay itself was not as fitting for this cultural context. Or that there are issues concerning the translation of some of these ideas into Spanish.

Because the data is not available, it's difficult to judge whether Spanish participants reacted to the material in a manner consistent with the sample of students from the U.S. Are mean reactions in the control group similar to the original study, for example?

Also, there is very little detail provided about the materials and procedure, so as to judge whether there were any inconsistencies from the original study.

In sum, I think it would be very hard to make any broad claims from the results of one replication of a particular study, in a different culture, with translated materials. Also, the particular study seems an odd choice, since it is not really the most central of the terror management hypotheses, and may be especially likely to have variable relevance as a function of cultural context. Finally, by linking the study to the general question of publication bias, it seems as if the authors may be drawing implications that are not quite appropriate on the basis of a single replication study (especially in light of the aforementioned issues).

Review form: Reviewer 3

Do you have any ethical concerns with this paper?

No

Have you any concerns about statistical analyses in this paper?

No

Recommendation?

Accept with minor revision

Comments to the Author(s)

This is a sound proposal to do a very worthwhile piece of replication research. The "Terror Management" type of priming findings require this kind of effort and the authors should be applauded for taking on this task.

Decision letter (RSOS-191114.R0)

07-Aug-2019

Dear Dr Vadillo,

The Editors assigned to your Stage 1 Replication submission ("Are we truly special and unique? A replication of Goldenberg et al. (2001)") have now received comments from reviewers. We would like you to revise your paper in accordance with the referee and editors suggestions which can be found below (not including confidential reports to the Editor). Please note this decision does not guarantee eventual acceptance.

Please submit a copy of your revised paper within three weeks (i.e. by the 29-Aug-2019). If deemed necessary by the Editors, your manuscript will be sent back to one or more of the original reviewers for assessment. If the original reviewers are not available we may invite new reviewers.

When submitting your revised manuscript, you must respond to the comments made by the referees and upload a file "Response to Referees" in the "File Upload" step. Please use this to document how you have responded to the comments, and the adjustments you have made. In order to expedite the processing of the revised manuscript, please be as specific as possible in your response.

Once again, thank you for submitting your manuscript to Royal Society Open Science and I look forward to receiving your revision. If you have any questions at all, please do not hesitate to get in touch. Full author guidelines may be found at <http://rsos.royalsocietypublishing.org/page/replication-studies#AuthorsGuidance>.

on behalf of Chris Chambers (Registered Reports Editor, Royal Society Open Science)
openscience@royalsociety.org

Editor Comments to Author (Professor Chris Chambers):

Three expert reviewers have now assessed the manuscript. Reviewers 1 and 3 are positive about the submission, judging that both Stage 1 primary criteria are met and recommending only minor revisions to clarify the power analysis, address deviations from the original study, and consider some additional analyses. Reviewer 2, however, is more critical, judging neither of the Stage 1 primary criteria to be met and recommending outright rejection. The reviewer raises concerns about the level of methodological detail in the manuscript and potential deviations from the original study, which are key points that will need to be thoroughly addressed in revision. Note that the concern raised by Reviewer 2 about the rationale for selecting the original study as a target for replication is not a relevant point for the RSOS Replication format, and therefore you need not respond to that concern in your response, although the issue of appropriate generalisation of conclusions beyond this specific replication becomes relevant if and when the manuscript is assessed at Stage 2.

Comments to Author:

Reviewer: 1

The authors propose to conduct a replication of a 2001 study about the mortality salience hypothesis. The original study found that a reminder of one's mortality produced a huge effect on subjects' evaluations of an essay emphasizing the distinction between humans and other animals, in keeping with the mortality-salience hypothesis.

My invitation to review this manuscript noted that the review process includes two stages. In stage 1, reviewers evaluate a manuscript that excludes the results, and, in stage 2, reviewers evaluate the completed manuscript. Notably, the journal explicitly allows for the submission of replication studies that have already been completed as long as the results are omitted from the Stage 1 manuscript.

I recommend that this replication proposal be accepted. The manuscript is well-written and clear, and the original study is sufficiently important (e.g., it appeared in top-tier journal and has been cited several hundred times). I have some minor comments and recommendations, which I'll include in my responses to the five review criteria listed in instructions for reviewers.

Primary Criterion #1: Whether the authors provide a sufficiently clear and detailed description of the methods ...

No concerns.

Primary Criterion #2: Whether the manuscript describes a sufficiently valid (i.e. close) and robust (e.g. statistically powerful) replication...

As for closeness of replication, the principal difference between the original study and replication study, as the authors note, is native language (original study in Colorado vs. replication study in Barcelona). Despite the salience of this difference, I don't think it's an issue because I cannot see why native language would moderate an effect related to one's sense of mortality. In fact, the two samples are not that different, as both samples consist of college students from a Western democracy consisting primarily of people who report beliefs aligning with Judeo-Christian principles. So, although I would ask that the authors list the English vs. Spanish difference in the final paper, I cannot see who anyone can argue that this difference explains why such a large effect would disappear (assuming that the effect disappears in the replication study).

As for statistical power, the intended sample size is only 44 subjects (22 per group), though the authors add that they will test more subjects if possible. The authors' power analysis indicates that 44 subjects provides statistical power of .95, but this analysis assumed an effect size of $d = 1.13$, which was observed in the original study. (The original study included 41 subjects.) Although the power calculation is appropriate, I think 44 is too few IF the replication study obtains an effect size that is greater than, say, $d = 0.20$. But if the effect size is trivial OR if the sample size is much larger than 44, I see no problems.

C. Secondary Criterion #1: The logic, rationale, and plausibility of the proposed hypotheses.

No concerns.

D. Secondary Criterion #2: The soundness of the methodology and analysis pipeline.

I would ask that the authors compare their obtained effect size to the original effect size (e.g., 95% confidence intervals).

E. Secondary Criterion #3: Whether the authors have considered sufficient outcome-neutral conditions (e.g. absence of floor or ceiling effects; positive controls; other quality checks)...

The possibility of floor or ceiling effects seems worthy of consideration. For that reason, perhaps the authors should report the M and SD for each of the six 9-point Likert questions that comprise the dependent measure (even though this was not done in the original study).

Reviewer: 2

Comments to the Author(s)

Review of "We are all Animals: A Failure to Replicate Goldenberg et al. (2001)"

To be frank, I am not able to adequately critique the first part of the analysis and comment on whether a publication bias may have influenced the effect size of the reviewed studies. I do not have the appropriate expertise to critique the analysis. I hope you will consult additional reviewers with respect to that issue.

But it does not seem clear to me how the efforts to replicate one study address the question about publication bias. Yet, it seems that is what the authors are intending to imply.

I also question the authors' decision with regard to the study they selected to test the mortality

salience effect, and the lack of consideration as to how the different cultural context and translation of content could affect the likelihood of replication. If the question is whether the tendency of mortality salience to promote efforts to protect one's worldview or self-esteem is in question, it seems important to determine that the worldview or self-esteem constituent is applicable to the population from which the sample is drawn.

Goldenberg et al. explicitly discuss the question of universality. They state that "even granting the proposition that all worldviews help people manage their terror of death, there may be cultural differences in the need to distance from animals." So really, the conclusion that can be drawn here is not that the results cast doubt on the reliability of mortality salience effects, but rather that the need to distance from animals as a means to defend against mortality salience may or may not be universal. But, it also may be that the essay itself was not as fitting for this cultural context. Or that there are issues concerning the translation of some of these ideas into Spanish.

Because the data is not available, it's difficult to judge whether Spanish participants reacted to the material in a manner consistent with the sample of students from the U.S. Are mean reactions in the control group similar to the original study, for example?

Also, there is very little detail provided about the materials and procedure, so as to judge whether there were any inconsistencies from the original study.

In sum, I think it would be very hard to make any broad claims from the results of one replication of a particular study, in a different culture, with translated materials. Also, the particular study seems an odd choice, since it is not really the most central of the terror management hypotheses, and may be especially likely to have variable relevance as a function of cultural context. Finally, by linking the study to the general question of publication bias, it seems as if the authors may be drawing implications that are not quite appropriate on the basis of a single replication study (especially in light of the aforementioned issues).

Reviewer: 3

Comments to the Author(s)

This is a sound proposal to do a very worthwhile piece of replication research. The "Terror Management" type of priming findings require this kind of effort and the authors should be applauded for taking on this task.

Author's Response to Decision Letter for (RSOS-191114.R0)

See Appendix A.

Decision letter (RSOS-191114.R1)

27-Sep-2019

Dear Dr Vadillo

On behalf of the Editor, I am pleased to inform you that your Manuscript RSOS-191114.R1

entitled "Are we truly special and unique? A replication of Goldenberg et al. (2001)" has been accepted in principle for publication in Royal Society Open Science.

You may now progress to Stage 2 and complete the study as approved.

Please note that you must now register your approved protocol on the Open Science Framework (<https://osf.io/rr>), using the 'Submit your approved Registered Report' option and then the 'Registered Report Protocol Preregistration' option. Please use the Registered Report option even though your article is being accepted as a Stage 1 Replication. Further into the registration process, in the Journal Title field enter 'Royal Society Open Science (Replication article type, Results-Blind track)'. Please note that a time-stamped, independent registration of the protocol is mandatory under journal policy, and manuscripts that do not conform to this requirement cannot be considered at Stage 2. The protocol should be registered unchanged from its current approved state. Please include a URL to the protocol in your Stage 2 manuscript, and because you submitted via the Results-Blind track please note in the manuscript that the pre-registration was performed after data analysis (e.g. 'This article received results-blind in-principle acceptance (IPA) at Royal Society Open Science. Following IPA, the accepted Stage 1 version of the manuscript, not including results and discussion, was preregistered on the OSF (URL). This preregistration was performed after data analysis.')

Following completion of your study, we invite you to resubmit your paper for peer review as a Stage 2 Replication. Please note that your manuscript can still be rejected for publication at Stage 2 if the Editors consider any of the following conditions to be met:

- The Introduction and methods deviated from the approved Stage 1 submission (required).
- The authors' conclusions were not considered justified given the data.

We encourage you to read the complete guidelines for authors concerning Stage 2 submissions at: <https://royalsocietypublishing.org/rsos/replication-studies#AuthorsGuidance>. Please especially note the requirements for data sharing and that withdrawing your manuscript will result in publication of a Withdrawn Registration.

Once again, thank you for submitting your manuscript to Royal Society Open Science and I look forward to receiving your Stage 2 submission. If you have any questions at all, please do not hesitate to get in touch. We look forward to hearing from you shortly with the anticipated submission date for your stage two manuscript.

Kind regards,
Andrew Dunn
Senior Publishing Editor
Royal Society Open Science
openscience@royalsociety.org

on behalf of Chris Chambers (Registered Reports Editor, Royal Society Open Science)
openscience@royalsociety.org

Author's Response to Decision Letter for (RSOS-191114.R1)

See Appendix B.

RSOS-191114.R2 (Revision)

Review form: Reviewer 1

Do you have any ethical concerns with this paper?

No

Have you any concerns about statistical analyses in this paper?

No

Recommendation?

Accept with minor revision

Comments to the Author(s)

The replication is sound. The authors conducted a high-powered replication and found an effect size of only $d = 0.09$, which is far less than the effect size of $d = 1.13$ found in the original (underpowered) study. I see no reason to doubt the results of the replication (e.g., no signs of floor or ceiling effect). I recommend publication.

I do have a few suggested edits (the authors appear to be non-native English speakers).

Abstract: no need to include title of original article

First paragraph: "...faith in one's..."

Page 4, para 1. "... statistical non-significance..."

Page 5, para 2 and elsewhere. No need to use quotation marks for "Study 1" and "Study 2"

Page 8, para 3. "... volunteers discovered..." I think this is some kind of copy/paste error.

Review form: Reviewer 2

Do you have any ethical concerns with this paper?

No

Have you any concerns about statistical analyses in this paper?

No

Recommendation?

Reject

Comments to the Author(s)

Are we truly special and unique? A replication of Goldenberg et al. (2001)

- Stage 2 Primary Criterion #1

There were changes from the manuscript I reviewed at Stage 1. The last two paragraphs of the Introduction deviate from the original Introduction. The authors changed their rationale for choosing this study since the Stage 1 review.

- Stage 2 Primary Criterion #2

My main criticism with the authors' conclusions concerns the choice of this study to test "the mortality salience effect." The authors of the original study explicitly state that "even granting the proposition that all worldviews help people manage their terror of death, there may be cultural differences in the need to distance from animals." So really, the conclusion that can be drawn here is not that the results cast doubt on the reliability of mortality salience effects, but rather that the need to distance from animals as a means to defend against mortality salience may or may not be universal. Or, it may be that distancing from animals is just somewhat more culturally relevant in the United States than Spain.

It also may be that the essay itself was not as fitting for this cultural context. Or that there are issues concerning the translation of some of these ideas into Spanish.

It is also not clear whether the student who presumably wrote the essay in the replication was from the same university as the participants, or a "local university" as in the original study. This may be relevant as the opinions would be coming from an ingroup and not an outgroup member, and may not have elicited as strong of a mortality salience reaction. I am sorry I did not notice this in the Stage 1 review, but it does, I would think, deserve mention.

These are all issues related to the question of whether the authors' conclusions are justified based on the data.

I agree with the criticism that failed replications can often be a result of a failure to take into consideration the contextual sensitivity of the processes involved in the effects. This is especially true for a study designed to measure a worldview as a defense.

In sum, I think it would be very hard to make any broad claims from the results of one replication of a particular study, in a different culture, with translated materials. Also, the particular study seems an odd choice, since it is not really the most central of the terror management hypotheses, and may be especially likely to have variable relevance as a function of cultural context. Finally, by linking the study to the general question of publication bias, it seems as if the authors may be drawing implications that are not quite appropriate on the basis of a single replication study (especially in light of the aforementioned issues).

Review form: Reviewer 3

Do you have any ethical concerns with this paper?

No

Have you any concerns about statistical analyses in this paper?

No

Recommendation?

Accept as is

Comments to the Author(s)

Seems very worthwhile and polished to me.

Decision letter (RSOS-191114.R2)

28-Oct-2019

Dear Dr Vadillo

On behalf of the Editor, I am pleased to inform you that your Stage 2 Replication submission RSOS-191114.R2 entitled "Are we truly special and unique? A replication of Goldenberg et al. (2001)" has been accepted for publication in Royal Society Open Science subject to minor revision in accordance with the referee suggestions. Please find the referees' comments at the end of this email.

The reviewers and Subject Editor have recommended publication, but also suggest some minor revisions to your manuscript. Therefore, I invite you to respond to the comments and revise your manuscript.

Please also ensure that all the below editorial sections are included where appropriate (a non-exhaustive example is included in an attachment):

- Ethics statement

- Data accessibility

<http://datadryad.org/submit?journalID=RSOS&manu=RSOS-191114.R2>

- Competing interests

- Authors' contributions

- Acknowledgements

- Funding statement

Because the schedule for publication is very tight, it is a condition of publication that you submit the revised version of your manuscript within 7 days (i.e. by the 05-Nov-2019). If you do not think you will be able to meet this date please let me know immediately.

- 1) A text file of the manuscript (tex, txt, rtf, docx or doc), references, tables (including captions) and figure captions. Do not upload a PDF as your "Main Document".
- 2) A separate electronic file of each figure (EPS or print-quality PDF preferred (either format should be produced directly from original creation package), or original software format)
- 3) Included a 100 word media summary of your paper when requested at submission. Please ensure you have entered correct contact details (email, institution and telephone) in your user account
- 4) Included the raw data to support the claims made in your paper. You can either include your data as electronic supplementary material or upload to a repository and include the relevant DOI within your manuscript
- 5) Included your supplementary files in a format you are happy with (no line numbers, Vancouver referencing, track changes removed etc) as these files will NOT be edited in production

Kind regards,
Lianne Parkhouse

Editorial Coordinator
 Royal Society Open Science
 openscience@royalsociety.org

on behalf of Professor Chris Chambers (Registered Reports Editor, Royal Society Open Science)
 openscience@royalsociety.org

Associate Editor Comments to Author (Professor Chris Chambers):

The three original reviewers who assessed the Stage 1 manuscript have now reviewed the completed Stage 2 submission. Reviewers 1 and 3 are broadly satisfied and recommend only minor stylistic revisions. Reviewer 2, however, remains very negative about the manuscript and recommends Rejection under both Stage 2 review criteria. Given the very precise criteria by which Replications are assessed, I have studied this review very carefully to assess the validity of the concerns.

The first concern of Reviewer 2 is potentially serious - that the authors deviated in their rationale between the Stage 1 and Stage 2 manuscript (specifically in the last two paragraphs of the Introduction). However, on specific inspection of the registered protocol (<https://osf.io/gts7c/>), and through comparison with both the Stage 1 manuscript that received IPA on the manuscript handling system, and the Stage 2 manuscript, I can find no evidence of such deviation. Instead, Reviewer 2 appears to have been comparing those sections of the Stage 2 manuscript with the *initial* version of the Stage 1 manuscript rather than with the registered Stage 1 manuscript, which was altered after the first round of Stage 1 review. In addition, these two paragraphs were revised specifically in response to Reviewer 2's comments at Stage 1. I therefore consider this specific criticism to be invalid and it can be disregarded by the authors.

Turning to Reviewer 2's second concern, in my reading most of the reviewer's concerns are already addressed in the Discussion, but there is one key point that appears not to be covered: "It is also not clear whether the student who presumably wrote the essay in the replication was from the same university as the participants, or a "local university" as in the original study. This may be relevant as the opinions would be coming from an ingroup and not at outgroup member, and may not have elicited as strong of a mortality salience reaction." As this is a comment concerning the design of the study, I agree with the reviewer that some mention of this issue should either be added to the Discussion or convincingly rebutted in response.

Reviewers' comments to Author:

Reviewer: 1
 Comments to the Author(s)

The replication is sound. The authors conducted a high-powered replication and found an effect size of only $d = 0.09$, which is far less than the effect size of $d = 1.13$ found in the original (underpowered) study. I see no reason to doubt the results of the replication (e.g., no signs of floor or ceiling effect). I recommend publication.

I do have a few suggested edits (the authors appear to be non-native English speakers).

Abstract: no need to include title of original article
 First paragraph: "...faith in one's..."
 Page 4, para 1. "... statistical non-significance..."

Page 5, para 2 and elsewhere. No need to use quotation marks for “Study 1” and “Study 2”
 Page 8, para 3. “... volunteers discovered...” I think this is some kind of copy/paste error.

Reviewer: 2

Comments to the Author(s)

Are we truly special and unique? A replication of Goldenberg et al. (2001)

- Stage 2 Primary Criterion #1

There were changes from the manuscript I reviewed at Stage 1. The last two paragraphs of the Introduction deviate from the original Introduction. The authors changed their rationale for choosing this study since the Stage 1 review.

- Stage 2 Primary Criterion #2

My main criticism with the authors’ conclusions concerns the choice of this study to test “the mortality salience effect.” The authors of the original study explicitly state that “even granting the proposition that all worldviews help people manage their terror of death, there may be cultural differences in the need to distance from animals.” So really, the conclusion that can be drawn here is not that the results cast doubt on the reliability of mortality salience effects, but rather that the need to distance from animals as a means to defend against mortality salience may or may not be universal. Or, it may be that distancing from animals is just somewhat more culturally relevant in the United States than Spain.

It also may be that the essay itself was not as fitting for this cultural context. Or that there are issues concerning the translation of some of these ideas into Spanish.

It is also not clear whether the student who presumably wrote the essay in the replication was from the same university as the participants, or a “local university” as in the original study. This may be relevant as the opinions would be coming from an ingroup and not an outgroup member, and may not have elicited as strong of a mortality salience reaction. I am sorry I did not notice this in the Stage 1 review, but it does, I would think, deserve mention.

These are all issues related to the question of whether the authors’ conclusions are justified based on the data.

I agree with the criticism that failed replications can often be a result of a failure to take into consideration the contextual sensitivity of the processes involved in the effects. This is especially true for a study designed to measure a worldview as a defense.

In sum, I think it would be very hard to make any broad claims from the results of one replication of a particular study, in a different culture, with translated materials. Also, the particular study seems an odd choice, since it is not really the most central of the terror management hypotheses, and may be especially likely to have variable relevance as a function of cultural context. Finally, by linking the study to the general question of publication bias, it seems as if the authors may be drawing implications that are not quite appropriate on the basis of a single replication study (especially in light of the aforementioned issues).

Reviewer: 3

Comments to the Author(s)

Seems very worthwhile and polished to me.

Author's Response to Decision Letter for (RSOS-191114.R2)

See Appendix C.

Decision letter (RSOS-191114.R3)

06-Nov-2019

Dear Dr Vadillo:

It is a pleasure to accept your Stage 2 Replication entitled "Are we truly special and unique? A replication of Goldenberg et al. (2001)" in its current form for publication in Royal Society Open Science.

Kind regards,
Anita Kristiansen
Editorial Coordinator
Royal Society Open Science
openscience@royalsociety.org

on behalf of Professor Chris Chambers (Subject Editor)
openscience@royalsociety.org

Appendix A

Comments made by the editor	Changes to the manuscript
Three expert reviewers have now assessed the manuscript. Reviewers 1 and 3 are positive about the submission, judging that both Stage 1 primary criteria are met and recommending only minor revisions to clarify the power analysis, address deviations from the original study, and consider some additional analyses. Reviewer 2, however, is more critical, judging neither of the Stage 1 primary criteria to be met and recommending outright rejection. The reviewer raises concerns about the level of methodological detail in the manuscript and potential deviations from the original study, which are key points that will need to be thoroughly addressed in revision. Note that the concern raised by Reviewer 2 about the rationale for selecting the original study as a target for replication is not a relevant point for the RSOS Replication format, and therefore you need not respond to that concern in your response, although the issue of appropriate generalisation of conclusions beyond this specific replication becomes relevant if and when the manuscript is assessed at Stage 2.	
Comments made by Reviewer 1	Changes to the manuscript
The authors propose to conduct a replication of a 2001 study about the mortality salience hypothesis. The original study found that a reminder of one's mortality produced a huge effect on subjects' evaluations of an essay emphasizing the distinction between humans and other animals, in keeping with the mortality-salience hypothesis. // My invitation to review this manuscript noted that the review process includes two stages. In stage 1, reviewers evaluate a manuscript that excludes the results, and, in stage 2, reviewers evaluate the completed manuscript. Notably, the journal explicitly allows for the submission of replication studies that have already been completed as long as the results are omitted from the Stage 1 manuscript. // I recommend that this replication proposal be accepted. The manuscript is well-written and clear, and the original study is sufficiently important (e.g., it appeared in top-tier journal and has been cited several hundred times). I have some minor comments and recommendations, which I'll include in my responses to the five review criteria listed in instructions for reviewers.	
Primary Criterion #1: Whether the authors provide a sufficiently clear and detailed description of the methods... No concerns.	
Primary Criterion #2: Whether the manuscript describes a sufficiently valid (i.e. close) and robust (e.g. statistically powerful) replication... As for closeness of replication, the principal difference between the original study and replication study, as the authors note, is native language (original study in Colorado vs. replication study in Barcelona). Despite the salience of this difference, I don't think it's an issue because I cannot see why native language would moderate an effect related to one's sense of mortality. In fact, the two samples are not that different, as both samples	In contrast to Reviewer 1, Reviewer 2 suggests that if we fail to replicate the original results this can be attributed to cultural factors or to details of the translation. These possibilities will be carefully considered in the General Discussion, at Stage 2. In any case, to facilitate the comparison of our materials and those of the original study, we now provide a link to the translated materials in Spanish, together with the original version in English. Of course, we understand that it will be difficult to assess the fidelity of our translation for non-Spanish speakers. But hopefully, Google Translator will provide a simple means to check the adequacy of the translation. It is perhaps worth noting

consist of college students from a Western democracy consisting primarily of people who report beliefs aligning with Judeo-Christian principles. So, although I would ask that the authors list the English vs. Spanish difference in the final paper, I cannot see who anyone can argue that this difference explains why such a large effect would disappear (assuming that the effect disappears in the replication study).	that our materials were back-translated to English by one of the authors (who at that time had not read the original version) and then both the original and the back-translated versions were checked by a bilingual collaborator, who failed to find any meaningful difference between them.
As for statistical power, the intended sample size is only 44 subjects (22 per group), though the authors add that they will test more subjects if possible. The authors' power analysis indicates that 44 subjects provides statistical power of .95, but this analysis assumed an effect size of $d = 1.13$, which was observed in the original study. (The original study included 41 subjects.) Although the power calculation is appropriate, I think 44 is too few IF the replication study obtains an effect size that is greater than, say, $d = 0.20$. But if the effect size is trivial OR if the sample size is much larger than 44, I see no problems.	In the previous version we mentioned that 44 participants is the minimum sample size that we would consider valid, although we would test more participants, if possible. In the present version, we clarify that, in practice, we expect to test at least 100 participants. Beyond this rough estimate, the exact number of difficult to anticipate, given the constraints of our participant recruiting system. But in any case, we hope that this will make clear that the sample will be substantially larger than in the original study.
C. Secondary Criterion #1: The logic, rationale, and plausibility of the proposed hypotheses. No concerns.	
D. Secondary Criterion #2: The soundness of the methodology and analysis pipeline. I would ask that the authors compare their obtained effect size to the original effect size (e.g., 95% confidence intervals).	We would like to thank R1 for this suggestion. We have included this analysis in the Results section.
E. Secondary Criterion #3: Whether the authors have considered sufficient outcome-neutral conditions (e.g. absence of floor or ceiling effects; positive controls; other quality checks)... The possibility of floor or ceiling effects seems worthy of consideration. For that reason, perhaps the authors should report the M and SD for each of the six 9-point Likert questions that comprise the dependent measure (even though this was not done in the original study).	In the present version of the manuscript, we include a Table where we will report the means and SDs for each item and group, together with the t, p, and effect size for the comparison of both groups on each item.
Comments made by Reviewer 2	Changes to the manuscript
To be frank, I am not able to adequately critique the first part of the analysis and comment on whether a publication bias may have influenced the effect size of the reviewed studies. I do not have the appropriate expertise to critique the analysis. I hope you will consult additional reviewers with respect to that issue. But it does not seem clear to me how the efforts to replicate one study address the question about publication bias. Yet, it seems that is what the authors are intending to imply.	See our response to this and other comments below.
I also question the authors' decision with regard to the study they selected to test the mortality salience effect, and the lack of consideration as to how the different cultural context and translation of content could affect the likelihood of replication. If the question is whether the tendency of mortality salience to	In the present version of the introduction, we address briefly the main reasons that led us to select this particularly study.

promote efforts to protect one's worldview or self-esteem is in question, it seems important to determine that the worldview or self-esteem constituent is applicable to the population from which the sample is drawn.	
Goldenberg et al. explicitly discuss the question of universality. They state that "even granting the proposition that all worldviews help people manage their terror of death, there may be cultural differences in the need to distance from animals." So really, the conclusion that can be drawn here is not that the results cast doubt on the reliability of mortality salience effects, but rather that the need to distance from animals as a means to defend against mortality salience may or may not be universal. But, it also may be that the essay itself was not as fitting for this cultural context. Or that there are issues concerning the translation of some of these ideas into Spanish. Because the data is not available, it's difficult to judge whether Spanish participants reacted to the material in a manner consistent with the sample of students from the U.S. Are mean reactions in the control group similar to the original study, for example?	See our response to this and other comments below.
Also, there is very little detail provided about the materials and procedure, so as to judge whether there were any inconsistencies from the original study.	As explained above, in the new version we tried to specify in more detail the procedure and we provide the full translation of the mortality salience manipulation and the "humans are unique" essay. If the reviewer still has doubts about any specific aspect of the experiment, we are happy to provide any further clarification.
In sum, I think it would be very hard to make any broad claims from the results of one replication of a particular study, in a different culture, with translated materials. Also, the particular study seems an odd choice, since it is not really the most central of the terror management hypotheses, and may be especially likely to have variable relevance as a function of cultural context. Finally, by linking the study to the general question of publication bias, it seems as if the authors may be drawing implications that are not quite appropriate on the basis of a single replication study (especially in light of the aforementioned issues).	Essentially, we agree with all the concerns posed by Reviewer 2 in this paragraph and above: Even if our study failed to replicate the original result, this would not mean that (a) all the effects explored in the MS literature are null or even that (b) the results of the original study are a null result. Of course, if the results of the replication depart from those of the original study, the cultural context will be a potential explanation, among others. We plan to address all these concerns at Stage 2, if our ms receives Stage 1 acceptance.
Comments made by Reviewer 3	Changes to the manuscript
This is a sound proposal to do a very worthwhile piece of replication research. The "Terror Management" type of priming findings require this kind of effort and the authors should be applauded for taking on this task.	We would like to thank Reviewer 3 for his/her kind words.

Appendix B

October 4th, 2019

Chris Chambers
Royal Society Open Science

Dear Prof Chambers,

Please find attached a revised version of our manuscript “Are we truly special and unique? A replication of Goldenberg et al. (2001)”, which we would like you to consider for publication in *Royal Society Open Science* as a replication study.

The present version of the manuscript is a Stage 2 submission where the Introduction, Method and Results sections are identical to the Stage 1 accepted protocol, except that placeholders for numerical values have been replaced by actual results. We have also added a Discussion section largely inspired by the comments of Reviewers 1-3 to the previous version. In addition, the current version of the manuscript includes an additional author, who had contributed to the design of materials and data collection.

We hope that you will find our revision to be suitable for publication in *Royal Society Open Science*.

Thank you very much for your time and attention in considering our revision.

Sincerely,

Miguel A. Vadillo
(on behalf of all the authors)
Departamento de Psicología Básica
Universidad Autónoma de Madrid
E-mail: miguel.vadillo@uam.es

Appendix C

Comments made by the editorial coordinator (Lianne Parkhouse)	Changes to the manuscript
Ethics statement: If your study uses humans or animals please include details of the ethical approval received, including the name of the committee that granted approval. For human studies please also detail whether informed consent was obtained. For field studies on animals please include details of all permissions, licences and/or approvals granted to carry out the fieldwork.	All the information related to ethical approval and informed consent has been sent to a new section called "Ethics statement". Note that this involves a minor change in the Methods section (which was part of Stage 1 accepted ms).
Data accessibility: It is a condition of publication that all supporting data are made available either as supplementary information or preferably in a suitable permanent repository. The data accessibility section should state where the article's supporting data can be accessed. This section should also include details, where possible of where to access other relevant research materials such as statistical tools, protocols, software etc can be accessed. If the data has been deposited in an external repository this section should list the database, accession number and link to the DOI for all data from the article that has been made publicly available. Data sets that have been deposited in an external repository and have a DOI should also be appropriately cited in the manuscript and included in the reference list. // If you wish to submit your supporting data or code to Dryad (http://datadryad.org/), or modify your current submission to dryad, please use the following link: http://datadryad.org/submit?journalID=RSOS&manu=RSOS-191114.R2	All the data are publicly available at the Open Science Framework https://osf.io/xt8kb/ as explained in the section Open practices statement.
Competing interests: Please declare any financial or non-financial competing interests, or state that you have no competing interests.	We have added a new section to the manuscript where we declare that we do not have any conflict of interest.
Authors' contributions: All submissions, other than those with a single author, must include an Authors' Contributions section which individually lists the specific contribution of each author. The list of Authors should meet all of the following criteria; 1) substantial contributions to conception and design, or acquisition of data, or analysis and interpretation of data; 2) drafting the article or revising it critically for important intellectual content; and 3) final approval of the version to be published. // All contributors who do not meet all of these criteria should be included in the acknowledgements. // We suggest the following format: AB carried out the molecular lab work, participated in data analysis, carried out sequence alignments, participated in the design of the study and drafted the manuscript; CD carried out the statistical analyses; EF collected field data; GH conceived of the study, designed the study, coordinated the study and helped draft the manuscript. All authors gave final approval for publication.	We have added this section to the ms.
Acknowledgements: Please acknowledge anyone who contributed to the study but did not meet the authorship criteria. Funding statement: Please list the source of funding for each author.	We have added a funding statement section, but no acknowledgements.

When uploading your revised files please make sure that you have:  1) A text file of the manuscript (tex, txt, rtf, docx or doc), references, tables (including captions) and figure captions. Do not upload a PDF as your "Main Document". 2) A separate electronic file of each figure (EPS or print-quality PDF preferred (either format should be produced directly from original creation package), or original software format) 3) Included a 100 word media summary of your paper when requested at submission. Please ensure you have entered correct contact details (email, institution and telephone) in your user account 4) Included the raw data to support the claims made in your paper. You can either include your data as electronic supplementary material or upload to a repository and include the relevant DOI within your manuscript 5) Included your supplementary files in a format you are happy with (no line numbers, Vancouver referencing, track changes removed etc) as these files will NOT be edited in production 	We have now included a "Media summary" on the first page.
Comments made by the associate editor (Chris Chambers)	Changes to the manuscript
The three original reviewers who assessed the Stage 1 manuscript have now reviewed the completed Stage 2 submission. Reviewers 1 and 3 are broadly satisfied and recommend only minor stylistic revisions. Reviewer 2, however, remains very negative about the manuscript and recommends Rejection under both Stage 2 review criteria. Given the very precise criteria by which Replications are assessed, I have studied this review very carefully to assess the validity of the concerns.	
The first concern of Reviewer 2 is potentially serious - that the authors deviated in their rationale between the Stage 1 and Stage 2 manuscript (specifically in the last two paragraphs of the Introduction). However, on specific inspection of the registered protocol (https://osf.io/gts7c/), and through comparison with both the Stage 1 manuscript that received IPA on the manuscript handling system, and the Stage 2 manuscript, I can find no evidence of such deviation. Instead, Reviewer 2 appears to have been comparing those sections of the Stage 2 manuscript with the *initial* version of the Stage 1 manuscript rather than with the registered Stage 1 manuscript, which was altered after the first round of Stage 1 review. In addition, these two paragraphs were revised specifically in response to Reviewer 2's comments at Stage 1. I therefore consider this specific criticism to be invalid and it can be disregarded by the authors.	
Turning to Reviewer 2's second concern, in my reading most of the reviewer's concerns are already addressed in the Discussion, but there is one key point that appears not to be covered: "It is also not clear whether the student who presumably wrote the essay in the replication was from the same university as the participants, or a "local university" as in the original study. This may be relevant as the opinions would be coming from an ingroup and not at outgroup member, and may not have elicited as strong of a mortality salience reaction." As this is a comment concerning the design of the study, I agree with the reviewer that some mention of this issue should	In the materials, the essay is presented as being written by an honors student in a "Catalan university", which, we think, preserves the ambiguity present in the original materials (i.e., "local university"). This is now clarified in the Method section.

either be added to the Discussion or convincingly rebutted in response.	
Comments made by Reviewer 1	Changes to the manuscript
The replication is sound. The authors conducted a high-powered replication and found an effect size of only $d = 0.09$, which is far less than the effect size of $d = 1.13$ found in the original (underpowered) study. I see no reason to doubt the results of the replication (e.g., no signs of floor or ceiling effect). I recommend publication.	
I do have a few suggested edits (the authors appear to be non-native English speakers). Abstract: no need to include title of original article First paragraph: "...faith in one's..." Page 4, para 1. "... statistical non-significance..." Page 5, para 2 and elsewhere. No need to use quotation marks for "Study 1" and "Study 2" Page 8, para 3. "... volunteers discovered..." I think this is some kind of copy/paste error.	We would like to thank Reviewer 1 for detecting these typos. All of them have been corrected in the current version.
Comments made by Reviewer 2	Changes to the manuscript
Stage 2 Primary Criterion #1: There were changes from the manuscript I reviewed at Stage 1. The last two paragraphs of the Introduction deviate from the original Introduction. The authors changed their rationale for choosing this study since the Stage 1 review.	As noted by the Associate Editor, the introduction/method is identical to the version that received IPA at Stage 1. We understand that this comment refers to changes included between the initial submission and the final Stage 1 accepted ms. Note that these paragraphs were included following the recommendations of Reviewer 2.
Stage 2 Primary Criterion #2: My main criticism with the authors' conclusions concerns the choice of this study to test "the mortality salience effect." The authors of the original study explicitly state that "even granting the proposition that all worldviews help people manage their terror of death, there may be cultural differences in the need to distance from animals." So really, the conclusion that can be drawn here is not that the results cast doubt on the reliability of mortality salience effects, but rather that the need to distance from animals as a means to defend against mortality salience may or may not be universal. Or, it may be that distancing from animals is just somewhat more culturally relevant in the United States than Spain. It also may be that the essay itself was not as fitting for this cultural context. Or that there are issues concerning the translation of some of these ideas into Spanish.	The current version of the manuscript already acknowledges that cultural factors might be responsible for the non-significant results. Note, however, that participants' responses in the control condition were identical to those of the original experiment. Nothing in our data supports the view that Spaniards are less motivated to distance themselves from animals than the population tested in the original study.
It is also not clear whether the student who presumably wrote the essay in the replication was from the same university as the participants, or a "local university" as in the original study. This may be relevant as the opinions would be coming from an ingroup and not an outgroup member, and may not have elicited as strong of a mortality salience reaction. I am sorry I did not notice this in the Stage 1 review, but it does, I would think, deserve mention.	The essay was presented as being written by an honors student in a "Catalan university". This is now mentioned in the Methods section.

These are all issues related to the question of whether the authors conclusions are justified based on the data.	
I agree with the criticism that failed replications can often be a result of a failure to take into consideration the contextual sensitivity if the processes involved in the effects. This is especially true for a study designed to measure a worldview as defense.	
In sum, I think it would be very hard to make any broad claims from the results of one replication of a particular study, in a different culture, with translated materials. Also, the particular study seems an odd choice, since it is not really the most central of the terror management hypotheses, and may be especially likely to have variable relevance as a function of cultural context.	It is beyond the scope of the present study to make any broad claims about the reliability of mortality salience effects. As we note in the final paragraph, we see this study as an initial step in the reevaluation of this literature.
Finally, by linking the study to the general question of publication bias, it seems as if the authors may be drawing implications that are not quite appropriate on the basis of a single replication study (especially in light of the aforementioned issues).	We honestly think that our treatment of publication bias in the introduction is extremely careful and balanced: "Of course, the previous analyses do not prove conclusively that these studies are influenced by publication or reporting biases And even if we could be completely sure that this set of studies is biased, that would not imply by any means that the effects explored in this literature are trivial or inexistent, although it would suggest that the effect sizes of the published record possibly overestimate the true effects of MS manipulations."
Comments made by Reviewer 3	Changes to the manuscript
Seems very worthwhile and polished to me.